# Additive Manufacturing of Polymer/Mg-Based Composites for Porous Tissue Scaffolds

**DOI:** 10.3390/polym14245460

**Published:** 2022-12-13

**Authors:** Fawad Ali, Sumama Nuthana Kalva, Muammer Koç

**Affiliations:** Division of Sustainable Development, College of Science and Engineering, Hamad bin Khalifa University, Qatar Foundation, Education City, Doha P.O. Box 34110, Qatar

**Keywords:** tissue scaffold, implant, Mg, 3DP, AM

## Abstract

Due to their commercial availability, superior processability, and biocompatibility, polymers are frequently used to build three-dimensional (3D) porous scaffolds. The main issues limiting the widespread clinical use of monophasic polymer scaffolds in the bone healing process are their inadequate mechanical strength and inappropriate biodegradation. Due to their mechanical strength and biocompatibility, metal-based scaffolds have been used for various bone regenerative applications. However, due to the mismatch in mechanical properties and nondegradability, they lack integration with the host tissues, resulting in the production of fiber tissue and the release of toxic ions, posing a risk to the durability of scaffolds. Due to their natural degradability in the body, Mg and its alloys increasingly attract attention for orthopedic and cardiovascular applications. Incorporating Mg micro-nano-scale particles into biodegradable polymers dramatically improves scaffolds and implants’ strength, biocompatibility, and biodegradability. Polymer biodegradable implants also improve the quality of life, particularly for an aging society, by eliminating the secondary surgery often needed to remove permanent implants and significantly reducing healthcare costs. This paper reviews the suitability of various biodegradable polymer/Mg composites for bone tissue scaffolds and then summarizes the current status and challenges of polymer/magnesium composite scaffolds. In addition, this paper reviews the potential use of 3D printing, which has a unique design capability for developing complex structures with fewer material waste at a faster rate, and with a personalized and on-site fabrication possibility.

## 1. Introduction

Biopolymers made from biological sources and with adjustable biodegradability are gaining popularity. Depending on where they came from, biopolymers are divided into natural and synthetic classes [1]. Biopolymers convey their payloads to human bodies, which undergo regulated degradation to form nontoxic byproducts [2]. This distinct characteristic makes tissue engineering viable for treating patients with damaged or missing organs and tissues [3]. Three-dimensional structures for repairing soft and hard tissues are typically created using natural and synthetic biopolymers. However, the poor mechanical properties and the slow degradation rate of biopolymer limit the use of biopolymer in tissue engineering [4].

The search for newer biodegradable and biocompatible materials for tissue engineering has recently increased due to the challenges driven mainly by secondary surgeries associated with the issues of the existing metal implants. Accidents, injuries, and the natural aging process cause bone degeneration, which requires an implant to restore its function [5,6]. According to a 2019 Australian Orthopaedic Association (AOA) report, the need for hip, knee, and shoulder surgical procedures has exponentially increased, reaching a total number of 459,265 hip, 658,596 knee, and 40,130 shoulder surgeries in 2018 [7]. Similarly, bone fracture costs only around USD 32 billion annually in the US. Over 3 million bone surgeries are conducted worldwide every year [8]. That is the main reason for the importance of highly efficient and cost-effective bone fracture treatment. The orthopedic implant costs USD 45.9 billion in 2017 and is expected to reach USD 66.6 billion by 2025, showing a compound annual growth of 4.7% [9]. The current demand for temporary implants is also soaring for scaffolds, cardiac stimulators, and cardiovascular stents [10,11,12,13,14,15]. Such a steep increase in orthopedic implants has necessitated the acceleration in the research and development for finding new materials and fabrication methods to satisfy the medical requirements of biodegradability, biocompatibility, and on-demand personalized design and manufacturing at a low cost.

Implants, acting as a replacement for natural bones, should imitate the bone in its physical and biological properties as much as possible, that is, strength, hardness, chemical properties, degradation rate, biocompatibility in the physiological environment, and so on [16,17]. There are mainly two types of implants: (1) permanent implants, which are required to perform the function of the bone for life with no immature failure and additional replacement needs, and (2) temporary implants, which are required to fulfill the functions of a bone until the tissues are healed [18,19]. Temporary implants stay inside the body, fixed to the bones, until the body part is fully recovered, and then are removed [20,21,22]. Currently, commercially available implants are mostly made of metals and metallic alloys, including titanium, stainless steel, and Co-Cr alloys, due to their high mechanical strength, biocompatibility, and good corrosion resistance properties [23,24,25,26,27,28,29,30,31]. However, due to the differences between the elastic modulus of these implant materials and natural bones, they impart a stress shielding effect [32,33,34,35]. Another problem with these materials is the requirement of a second surgery to remove these implants after the healing of the fracture, causing pain to the patient. Such a process not only adds pain, discomfort, and risks to the patient but also costs about 30% of healthcare expenditures [36,37]. Studies have found that failing to remove these implants can cause severe allergic problems due to ion accumulation, causing osteolysis [36,37].

Therefore, using biodegradable materials that the body may dissolve is significant for these implants. Such biodegradable implants will replace the natural bone until its recovery without any complications. Owing to its low density (1.8 g/cm^3^), high strength-to-weight ratio, compressive yield strength, and comparable elastic modulus to natural bone, magnesium (Mg) and its alloys (with some bioactive materials) can be a perfect choice over the existing metal implants [38,39,40]. Mg, a crucial component of natural bone formation and the body’s fourth-most principal cation, has good biocompatibility and is among the ideal materials for temporary implants. On average, a healthy adult weighing 70 kg is known to have 21 to 35 g of stored magnesium [41,42]. Twenty percent of the available amount of Mg is kept in bones, followed by 35–40% in tissues and ligaments and 1% in bodily fluid [43]. In addition, over 300 enzyme-related processes rely on the availability of Mg, either directly or indirectly [44,45]. Mg has the second-highest daily permissible intake (420 mg) of all the nutritionally necessary substances [43]. Additionally, the byproducts of the breakdown of magnesium and its alloys can be safely taken by macrophages and eliminated through the urine without endangering physiological function [44].

Despite having several valuable qualities, the unpredictable deterioration rate of Mg-based implants in the physiological environment prevents their commercialization as temporary implants. Mg-based implants deteriorate significantly more quickly in the body fluid than the conventional aqueous solution due to high chloride ion concentration (96–106 mEq/L) and a pH of roughly 7.4–7.6 [46]. Additionally, the accelerated rate of gas bubble creation harms the patient’s health. The implants become particularly susceptible to unexpected failure and lose the necessary mechanical integrity to support the load during this deterioration phase. As a result, creating Mg-based implants with a controlled degradation rate to prolong the bone-required repair duration (between 24 and 32 weeks) is essential.

Tissue engineering scaffolds frequently use polymeric materials, such as polylactic acid (PLA), poly(lactide-co-glycolic acid) (PLGA), and polycaprolactone (PCL), due to their simple processing by 3D printing [47,48], biocompatibility, osteoinductivity, and negligible inflammatory response. Despite such appealing properties of the polymers that could be utilized as a 3D porous scaffold, inadequate mechanical strength and an improper degradation rate call for additional study in this area [49]. Metal particles as a reinforcement to a polymer matrix is an effective strategy for improving polymers’ biological behavior and mechanical properties. Due to its magnificent properties in biomedical applications, Mg can be an excellent addition to polymer materials, further enhancing biodegradable polymers’ mechanical and biological properties.

Polymer/Mg composite scaffolds must be designed with high dimensional accuracy and reproducible. Furthermore, these scaffolds should be designed according to the patient-specific anatomic requirements. 3D printing technology, as one of the advanced manufacturing techniques, offers high dimensional accuracy and excellent reproducibility with low-cost manufacturing. It can produce customized or personalized structures based on patient-specific anatomic data. Three-dimensional printing controls porosity and surface topology, determining the final implant’s mechanical, chemical, and biodegradation properties. Three-dimensional printing is known to be an easy, fast, and on-site manufacturing process, which can fabricate implants closely resembling native bone properties. This paper elucidates the applications of polymer/Mg composites for biomedical applications. This review summarizes the polymer/Mg composites for biomedical applications, their manufacturing routes using modern 3D printing techniques, and the challenges.

## 2. Suitability of Polymer/Mg Composites for Biomedical Applications

Tissue scaffolds for temporary implants must fulfill several essential functions. Magnesium, in combination with some polymers, meets many of these properties. The required properties undoubtedly change depending on the intended use and requirement. However, while we primarily concentrate on transient in orthopedic fixtures, biocompatibility, good mechanical properties, natural degradability, and osteogenesis are the most desired qualities. In-depth reviews of these composite designing, material processing, and fabrication techniques used to enhance the Mg/polymer composite qualities are provided in the following sections.

### 2.1. Biocompatibility

Biocompatibility and nontoxicity are the two most crucial requirements for every implant material. Soon after a foreign object is inserted into a human body, interactions between the body tissues and implanted material start to happen. These reactions determine whether the body will accept the implant [50]. Biocompatibility for permanent implants depends on how well the newly formed tissue fuses with the implant surface. Contrastingly, temporary implants are intended to stabilize broken bone while it heals before degrading naturally in the body over time [8,51]. It is pretty concerning whenever a physiological component starts by negatively impacting the breakdown product. Consequently, nontoxic materials should be used for temporary implants.

The 3D porous bone scaffolds’ cellular activity is significantly affected by adding Mg particles into polymer matrices, such as PLA, PCL, and PLGA, during 3D printing. The presence of Mg is advantageous for cell growth, proliferation, and differentiation because composite scaffolds have more hydrophilicity, more bioactive area on their surface, and a moderately alkaline pH (pH 10), which is a favorable microenvironment for cell growth [52]. The amount of Mg in the polymer is a crucial element that needs to be tuned since an increase in the pH beyond 10 (pH > 10) harms cellular activity. Excess Mg (greater than 10 wt% (8.9 vol%) in PLGA and five wt% in PCL) has been reported to hurt cytocompatibility [53,54]. Figure 1 shows suitable cell attachment and proliferation on PCL/Mg composite scaffolds created by the FDM process. Cell viability (green = viable cells, red = dead cells) is markedly increased by adding five wt% Mg particles and decreased by further augmentation, as revealed in CLSM images (Figure 1a–f). Similarly, SEM photos demonstrate that cells in the PCL/Mg scaffold stretch, freely expand, and initiate filopodia, establishing a better cellular adherence to composite scaffolds. However, cells barely attached to the surface on the monophasic PCL scaffold with inappropriate conditions are unacceptable (Figure 1a–f) [53].

### 2.2. Mechanical Integrity

The material suitability to be employed as an orthopedic implant for a particular purpose is highly dependent on its mechanical qualities. Mechanical properties, such as elastic modulus, tensile strength, fatigue strength, hardness, and elongation, are among the rates given top priority. To prevent the “stress shielding effect”, the elastic modulus of biomedical implants should ideally have a similar value to the natural bone [32,33]. Biopolymers, owing to their excellent biocompatibility and biodegradability, have a strong possibility for scaffolds used in bone tissue engineering. However, they have poor mechanical qualities. This point of view claims that measures have been taken, such as customizing the pore structure and shape to overcome the substandard attribute qualities of the polymer 3D-printed scaffolds [55], modifying printing parameters [56], and incorporating other phases [57,58]. Among these efforts, the use of metallic powders as fillers is the most promising [59,60]. Similarly, adding Mg particles into the PLLA matrix demonstrated good mechanical properties (compressive strength and modulus of 5 wt% (3.6 vol%) Mg particles to the PLLA matrix considerably enhanced the compressive strength and modulus of 3D porous scaffolds by 114.5% and 85.7%, respectively). Adding more magnesium can negatively affect the mechanical characteristics of the porous composite scaffold made using the SLS process because of the particle agglomeration [61]. Alizadeh et al. reported a magnesium-reinforced PCL matrix [12]. The correct quantity of magnesium reinforcement strengthens mechanical capabilities, but too much magnesium weakens support due to the regional enhancement of magnesium microparticles in the PCL matrix. The compressive modulus of PCL/Mg scaffolds is summarized in Figure 2 [62,63,64,65,66,67,68,69,70,71,72,73,74,75,76]. Comparatively, PCL-based scaffolds have a lower compressive modulus than Mg-based scaffolds, which helps to limit the stress shielding effect. Additionally, Mg/PCL scaffolds created using 3DP had a greater compressive modulus than those made through electrospinning and salt leaching at a comparable porosity [63,64]. This scaffold’s mechanical attributes were like those of human cancellous bone, reducing the stress shielding effect. Ma et al. [77] observed an increase in compressive modulus by adding 20 wt% (18.1 vol%) Mg in the 3D porous PLGA scaffold. This enhancement was primarily ascribed to the PLGA/Mg composite scaffold’s increased load-bearing capability because Mg fillers have better strength and modulus.

### 2.3. Biodegradation

The relationship between the synthetic tissue scaffolds’ biodegradability or degradation rate and the rate at which bones mend is crucial. Because the hydrolytic breakdown pathway causes the polymeric chains to separate into monomers and water-solvable oligomers, biopolymers, including PLA, PCL, and PLGA, disintegrate in the physiological environment [49]. The total breakdown of the mentioned polymers in the body would take several years, however [78]. One of the significant problems with biodegradable polymer-based scaffolds is their sluggish disintegration rate, which needs more research. On the other hand, Mg, a very suitable metal with good biocompatibility, has a very high deposition rate. Therefore, incorporating Mg particles into these polymers in a specific ratio can have a controlled degradation rate matching that of the natural bone growth rate and having a good match of the mechanical properties similar to the natural bone. The incorporation of metallic powders into polymers is an effective strategy among the various methods used to increase the degradation of polymers without sacrificing biocompatibility, such as copolymerization [79], blending [80], and surface modification [81]. This is due to the outstanding ability to increase the strength and degradation rate simultaneously [61].

### 2.4. Osteogenic and Angiogenic Characteristics

The process of osteogenesis involves creating new tissues to mend a broken bone. It is a crucial necessity for temporary implants since the damaged bone needs to heal with the help of newly formed tissues and cells before implant disintegration [82,83]. The osteogenic activity of the synthetic polymers used in the 3D printing of 3D porous scaffolds needs to be improved because they are nonbioactive [84]. A promising strategy in this area has been the fabrication of polymer/metal composite scaffolds and introducing metallic fillers. Due to its inherent ability to promote osteogenesis, Mg is highly preferable compared with other bioactive materials. This suggests that Mg positively affects the osteogenic differentiation of 3D composite porous scaffolds and bone mineralization. According to reports, adding Mg to monophasic PLGA [54] and PCL [53] bone scaffolds remarkably increases the ability of genes relevant to osteoblast development and mineralization to express. The enhanced expression of neuronal calcitonin gene-related polypeptide (CGRP) by an Mg filler is suggested to be the cause of the polymer/Mg composite scaffolds’ robust bone formation.

Jing Bai et al. [53] have shown the general research methodology for Mg/PCL scaffolds for bone repair, as shown in Figure 3. The homogeneous distribution of Mg particles in PCL scaffolds is made possible by combining mixing, blending, and 3D printing. Scaffolds made of a Mg/PCL composite show good overall qualities and in vitro and in vivo responses. The Mg particles slowly degrade with a local pH increase in in-vitro degradation studies.

Mg/PCL scaffolds encourage cell growth, proliferation, vascularization, and bone formation in in vitro biological investigation.

## 3. Mechanism of Properties’ Enhancement of Biodegradable Polymers with Mg Addition

Polymers are the fastest-growing category in the biomedical market share compared with other materials in the last decade. This astounding growth is attributed to better biocompatibility, bioinertness, and the ease of fabrication of biopolymers compared with metals for biomedical applications. One of the significant problems with biodegradable polymer-based scaffolds is their sluggish disintegration rate [78] and lower mechanical strength, which needs more research to be addressed. Various techniques have been used to increase the degradation behavior of polymers without sacrificing their biocompatibility, such as copolymerization [79], blending [80], and surface modification [81]. Mixing metallic powders with polymers is a promising strategy among various methods because metallic powders have the unique capacity to simultaneously increase the strength and biodegradation at the same time [61]. Magnificent developments in Mg-based biomaterials research in the current decade of the 21st century have tremendously facilitated their use in biomedical applications. Owing to its suitable strength and higher biodegradability rate in a physiological environment, Mg can be an excellent addition to the biopolymer for adjusting the degradation rate of the polymer and enhancing its mechanical properties.

The polymer/Mg composite scaffolds’ degradation process accelerated with Mg is shown in Figure 4. The degradation process of the polymer/Mg composite is explained by Shui et al. [61] for a PLLA/Mg-based composite. As shown in Figure 4, the PLLA/Mg scaffolds absorbed more water due to the PBS solution invading the Mg particles and PLLA matrix interface with ease, whereas it was challenging to wet the PLLA scaffold due to its hydrophobicity [85]. Then two responses would happen at once. In one sense, according to Equation (1), PLLA would hydrolyze by rupturing the ester linkages to yield acidic products:PLLA ➔ R_1_—COOH + R_2_—OH(1)

On the other hand, Mg degrades to Mg(OH)_2_ and H_2_, as shown below:Mg + H_2_O ➔ Mg(OH)_2_ + H_2_ ↑(2)

The acid products produced in Equation (1) could then be consumed Mg(OH)_2_, built in Equation (2), promoting hydrolysis (Equation (3)) [45]. In contrast, the alkaline products of PLLA might consume the acid degradation products, promoting the breakdown of Mg.
Mg(OH_2_) + 2R1—COOH ➔ (R1COO)_2_Mg + 2H_2_O(3)


As a result, more Mg(OH)_2_ and PLLA acid degradation products were created. These compounds were then consumed, which formed positive feedback on the breakdown of the scaffolds [86].

**Figure 4 polymers-14-05460-f004:**
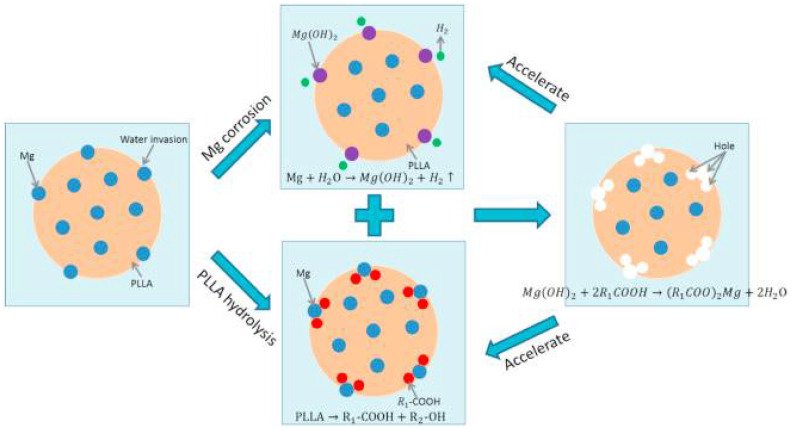
The mechanism of accelerated degradation of polymers by Mg incorporation [61].

Due to the inherent qualities of metals nano/microparticles, such as their vast surface area and high modulus, metal nano/microparticles frequently enhance the strength of the polymers. Additionally, the accomplishment of many mechanical characteristics is facilitated by the creation of a solid interfacial/interphase bonding between the polymer matrix and metal particles [87]. The correct quantity of magnesium reinforcement strengthens mechanical capabilities, but too much magnesium weakens support due to the local augmentation of magnesium microparticles in the polymer matrix. Interfacial adhesion and effective stress transfer are the most crucial element that increases the strength of polymer/metal composite materials [88]. Many studies have reported an improvement in the mechanical properties of biodegradable polymers by the addition of Mg particles [53,61,89,90,91].

## 4. Conventional Techniques for Biopolymer and Polymer/Mg Composites

The characteristics of the resulting scaffolds are determined by the production method and material choices. Even though various techniques have been tried to make the scaffolds more porous and prevent cellular ingrowth, none of them have successfully created scaffolds thick enough to qualify as 3D scaffolds [92]. The numerous traditional methods used to create porous magnesium polymer composite scaffolds for biomedical purposes are briefly described in the following section.

*Self-assembly*: Self-assembly is the ability to assemble components into patterns and structures that can be used to create designs that mimic the ECM for bone tissue creation [93]. A trigger from the outside, such as a change in pH or temperature, might start the assembling mechanism in motion [94,95]. To create HA/Coll biohybrid composites, an approach inspired by biological mineralization was used [96]. The magnesium doping of the apatite phase nucleating on collagen caused the final composite (MgHA/Coll) to exhibit the physicochemical, structural, and morphological features typical of a newly formed natural bone. Intrinsic control mechanisms, including chemistry, morphology, and spatial distribution of the mineral phase, occur throughout the artificial mineralization process, much like they do during the natural biomineralization process. Despite its many benefits, this method produces scaffolds with poor mechanical strength and endocytosis-risking broken fibers. Additionally, the mechanisms driving the self-assembly are more intricate, necessitating a complex and meticulous experimental design. In addition to these restrictions, the high cost of synthesis prevents them from being used in regenerative medicine and tissue engineering [97].

*Thermally induced phase separation*: A homogeneous polymer solution experiences a thermal energy differential during quenching, which starts the void formation. High temperatures solubilize the polymer, and then the temperature is rapidly lowered to trigger phase separation. The solution then separates into a solvent-free phase and a phase free of polymers through vigorous fluid de-blending. This method can adjust the polymer characteristics, solvents, and working temperature to alter the scaffold’s microstructure [98]. This method gives better mechanical qualities with a controlled porous structure and can be compatible with many other manufacturing procedures. Chitosan-magnesium-based composite scaffolds were successfully synthesized by Adhikari et al. through thermally induced phase separation [99]. The platforms had consistent porosity with 50–250 m–sized pores that were tightly linked. Elastic moduli of up to 5 MPa and compressive strengths of up to 400 kPa were found. The in vitro testing revealed that the scaffolds held on to their original three-dimensional frameworks and were unharmed. When exposed to these scaffolds, the 3T3 fibroblast and osteoblast cells showed no cytotoxicity.

*Melt molding*: Melt-based fabrication techniques have their roots in traditional polymer manufacturing methods. To create porous material, pore-generating techniques can be used [100]. It is common practice to combine water-soluble salts with the polymer during molding and dissolve them in water afterward to produce porous structures. The benefits of this approach include the avoidance of hazardous solvents and the ability to regulate pore size by utilizing porogens of the appropriate size [100]. This method exhibits significant advantages for creating ECM-mimetic tissue regeneration studies by integrating techniques including particle leaching, gas foaming, and the usage of porogens. Dutta et al. created porosity in magnesium scaffolds by using spherical naphthalene particles as the porogen in a powder metallurgy process [101]. Porogen was eliminated by heating at 120 °C for 24 h; then the material was sintered in an argon environment for 2 h at 550 °C. The scanning data show that scaffolds have connected porous structures with a pore size of around 60 µm. The scaffolds’ compressive strength was determined to be between 24 and 184 MPa, and it dropped as the porogen concentration rose. A study of in vitro degradation in phosphate-buffered saline (PBS) revealed that the porosity content of the scaffold controlled how quickly it degraded.

*Solvent casting:* One of two methods for creating scaffolds via solvent casting is based on the evaporation feature of particular solvents. One method involves dipping a mold into the polymer solution and giving it enough time to draw off the solvent so that the polymeric membrane layer can form. The alternative method involves pouring the polymer solution into a mold and giving it enough time to evaporate so that a layer of the membrane adheres to the mold. This method requires no specialist equipment, is relatively straightforward to use, is affordable, and has minimal effects on degrading behavior. However, it employs highly hazardous solvents that can denature proteins and other integrated compounds, but this can be avoided by letting the scaffold completely dry out utilizing a vacuum procedure. Some researchers have coupled solvent casting with other approaches, such as salt leaching, to increase the scaffold’s features while avoiding the solvent casting’s drawback approaches, such as salt leaching. This method, meanwhile, is only effective when building very thin scaffolds. Solvent casting and salt leaching procedures were used to create porous metallic magnesium/PLGA scaffolds [102]. It was observed that adding various amounts of magnesium to PLGA scaffolds improved their compressive strength and modulus while also creating a porous structure ideal for cell infiltration. Additionally, a pH buffering effect and long-term magnesium relation of a 10-week degradation trial were obtained by reacting basic-degrading magnesium with acidic-degrading PLGA. Micro-CT and histological examination of the magnesium/PLGA scaffolds revealed that they were safer and more efficient in preserving bone height than empty controls. This study showed that 3D magnesium/PLGA composite scaffolds show potentially promising applications for orthopedic bone regeneration.

*S/O/W emulsion method*: Solid particles can be dissolved in an oil phase to create solid-oil dispersion, which can then be added to the water phase to create a second emulsion to entrap solid particles. This is known as the “solid-in-oil-in-water (s-o-w) double-emulsion method”. The s-o-w emulsion technique is utilized to get around some issues that are typically present when entrapping hydrophilic particles using the w-o-w emulsion system, such as stability, encapsulation efficiency, burst release, and agitation stress. The s-o-w emulsion is more stable due to the adsorption of solid particles at the fluid–solid interface, which enhances the development of interfacial films. With reduced damage and greater entrapment efficiency, this technique can successfully entrap delicate materials. The scaffolds used today for bone tissue creation still do not have enough osteogenic potential to stimulate bone regeneration effectively. To achieve their synergistic effect on inducing osteogenesis, biodegradable microsphere-type scaffolds are created to accomplish the dual-controlled release of drugs and a bioactive ion (i.e., Mg^2+^). Poly(lactide-co-glycolide) (PLGA) microspheres coembedded with MgO and MgCO3 were used to create biodegradable microspheres (PMG) using s-o-w emulsion method [103]. By adjusting the MgO/MgCO_3_ ratios, the PMG microspheres displayed a regulated release of Mg^2+^. Higher MgO fractions showed faster release with higher initial Mg^2+^ concentrations, while higher MgCO_3_ fractions showed long-term sustained release with lower Mg^2+^ concentrations. These results suggest that regulated Mg^2+^ release from bioresorbable microspheres was effective in repairing bone defects and showed promising potential for future in vivo uses. In a separate study, it was discovered that the coinclusion of icariin had additional effects on stimulating the formation of a new bone. These findings indicate a promising method for healing severe bone injuries by creating a dual-release system [104].

*Decellularization*: Decellularization preserves the biological activity, biochemical composition, 3D organization, and integrity of the native ECM while clearing tissues and organs of cells and debris [105]. Due to the absence of foreign cells in the decellularized constructs, there is little possibility of immunological rejection [106]. Treatments using physical, chemical, and enzymatic processes can be used to decellularize in various ways. A technique that delivers locally while requiring less intrusive intervention can be created by processing several matrices to create injectable membranes. It receives acceptance in additional disciplines, such as drug testing and stem cell research [107]. Lih et al. created a scaffold that is bioinspired and effectively stimulates the regeneration of renal tissue. The decellularized renal extracellular matrix, magnesium hydroxide, and poly(lactide-co-glycolide) (PLGA) were used to create the bioinspired scaffold (ECM). By neutralizing the acidic environment created by PLGA degradation products, Mg(OH)_2_ prevented materials-induced inflammatory reactions, and the acellular ECM assisted in restoring the biological function of kidney tissues. In a partly nephrectomized mouse model, the PLGA/ECM/Mg(OH)_2_ scaffold promoted renal glomerular tissue regeneration with a minimal inflammatory response [108].

*Electrospinning*: This method uses electric voltage to create a three-dimensional structure with fibers that range in size from nanometers to micrometers and have a larger surface area [109]. This technique makes use of numerous natural and synthetic polymers, including PCL, gelatin, collagen, polyvinyl alcohol (PVA), and polycaprolactone (PCL) [110]. Pore diameters in typical electrospun scaffolds fall between 5 and 150 m. These scaffolds offer nanoscale fiber structures with linked pores to resemble native ECM, demonstrating the potential to create functional tissues. This approach has many benefits, including flexibility, nonobtrusive, and temperature independence [111]. The construction of scaffolds for tissue-specific activities is facilitated by imparting bioactivity in addition to structural features [101,102,103]. By adding magnesium oxide (MgO) nanoparticles to scaffolds made of silk fibroin and polycaprolactone (SF/PCL) blend, a magnesium-containing membrane was created by electrospun tissue engineering [112,113,114,115]. Investigations were conducted on the kinetics of Mg^2+^ release, the impact of magnesium on scaffold architecture, and cellular behavior. The generated nanofibrous scaffolds with Mg functionality exhibited regulated Mg^2+^ release, good biocompatibility, and osteogenic potential. Many weeks after surgery, a rat with a calvarial defect that had been in vivo implanted with an electrospun nanofibrous membrane containing magnesium had a considerable improvement in bone regeneration. Mg/(Pcomposite)-blended nanofiber scaffolds with different ratios were studied by Li et al. [116]. The degradation tests revealed that pure P had better mechanical properties and biocompatibility than Mg/P(LLA-CL)-blended nanofibers P(LLA-CL). Additionally, when Mg/P(LLA-CL)-composite nanofibers degraded, fibers’ weight loss was hastened by the addition of Mg, and the pH levels of the environment were raised. In another study, the electrospinning technique was used to fully manufacture magnesium/PCL membranes that were doped with magnesium particles. The matan excellent had a good porosity structure, and the magnesium/PCL electrospun membranes were observed to cause the hDPSCs to differentiate into osteoblasts and exhibit good biocompatibility [117].

## 5. Three-Dimensional Printing Techniques for Biopolymer and Polymer/Mg Composites

Even though the conventional scaffold preparation techniques discussed in the previous section offer great benefits, the most practical method to create 3D porous scaffolds with intricate pore structures is thought to be additive manufacturing technology. Three-dimensional printing is a technique that creates layer-by-layer 3D haptic physical models based on CAD models [118]. Polymer composites have been made using a variety of printing techniques: fused deposition modeling (FDM), low-temperature deposition manufacturing (LTDM), and selective laser sintering (SLS) [119]. Due to its excellent resolution, ease of use, and material accessibility, FDM, the most popular 3D printing technique, can build polymer and polymer-based composite scaffolds well [120,121,122,123,124,125,126]. Another approach used to create porous and sensitive scaffolds is LMD. LMD is a procedure that combines 3D printing with phase separation techniques to create two classes of macropore (100–850 m) and micropore (10 m) structures, both of which are necessary for cellular activity [127]. With a proper pore structure and improved mechanical qualities, SLS can produce a wide range of bone tissue engineering scaffolds that have been successful in the production of polymer/metal porous scaffolds [61].

In all the 3D printing techniques, the process parameters and materials properties are fundamental in the determining the final product quality. The printed structures are prone to failure fractures or internal defects [128]. Therefore, optimized printing parameters are important to avoid such defects and microcracks. The quality of printed parts depends on various underlying physical phenomena during printing. The bond formation between two layers includes surface contacting, neck growth, and molecular diffusion. Additionally, the difference in temperature profiles between adjacent layers during the solidification causes shrinkage to stress and distortion of the printed part [129]. In order to improve the properties of printed parts, literature studies have concentrated on creating composite material systems by adding various fillers to the base polymer [130]. In comparison with items printed from unreinforced polymers, products manufactured employing these composites have better mechanical and biological properties. These properties highly depend on the reinforcement particle size, shape, concentration, and distribution. A higher concentration of particles usually decreases the properties due to agglomerate formation [126].

### 5.1. Fused Deposition Modeling (FDM)

Crump created FDM in 1989, today the most popular 3D printing method for producing intricate geometrical pieces on the market [36]. In this method, a continuous filament of the suggested manufacturing material is fed into the heated nozzle. The heated semisolid filament is extruded, deposited, and then solidified to form the fundamental part, according to data from a 3D model (a computer-aided design or CAD file). In FDM, filaments melt into a semiliquid state at the nozzle and are extruded one layer at a time onto the build platform, where the layers combine and subsequently solidify to form a finished object [122]. The quality of printed components can be changed by modifying printing parameters, such as layer thickness, printing orientation, raster width, raster angle, and air gap. Figure 5 depicts the printing process.

Due to its exceptional ability to produce complex shapes with excellent resolution, use of inexpensive tools or molds, ease of use, compatibility with most thermoplastic polymers such as PLA and PCL, minimal production of waste and pollution, and numerous other benefits, the FDM process has emerged as one of the most important 3D printing processes to be used in biomedical applications [131,132,133]. However, the entirely polymeric printed pieces’ poor mechanical qualities restrict their uses to some extent in the load-bearing tissue engineering scaffolds [134].

The FDM process has prepared several polymer/Mg composite 3D porous scaffolds. In this method, the beginning materials of polymers in the form of pellets and metal powder are mechanically or melt-mixed before being added to an extruder to create the polymer/metal composite filament. To create the composite porous scaffolds with the necessary geometry and dimensions, a 3D desktop printer using the composite filament is used. The process variables used to build the composite porous scaffolds are also summarized in Table 1. The FDM 3D printing technique can create polymer/Mg composite porous tissue scaffolds with intricate geometries and interconnected pore structures. FDM printers provide benefits, including low-cost, high-speed printing and ease of use. The ability to enable the simultaneous deposition of many materials is another benefit of FDM printing. FDM printers can be configured with several extrusion nozzles loaded with various materials, allowing for printing multifunctional parts with a specific composition.

### 5.2. Selective Laser Sintering (SLS)

SLS is a laser-based 3DP technology frequently used to deposit 3D porous bone scaffolds with complicated pore patterns. For a predetermined route scan, a laser beam selectively heats and sinters a tiny layer of powder in this manner. When finished, a fresh layer of powder is spread out over the lifting platform before being lowered. This procedure keeps adding layers until the 3D porous scaffold is fully constructed entirely. Typically, the unbonded powder is removed using an air compressor [140,141]. A schematic illustration of this process is presented in Figure 6. SLS is a potential technique for manufacturing bone tissue scaffolds that resemble the bone’s anatomical geometry and include functionally graded porosity structures [142]. Furthermore, the polymeric-metal composite scaffolds created using this technique resemble in mechanical essential characteristics those of a human trabecular bone [143].

SLS has been used to create polymer/metal composite scaffolds in the 3D porous form, such as the poly-lactic acid (PLLA)/Mg scaffold [61]. The desired ratio of polymer and metal particles is mechanically blended or can be disseminated and mixed into a solvent to create a homogenous solution before the SLS process. The answer is dried in an oven at temperatures below 100 °C to produce the polymer/metal particles. The 3D porous polymer/metal scaffold is built using SLS technology, which involves treating the resultant composite powder with a laser beam. This has been accomplished using a CO_2_ laser with a 2.3 W power and a 120 mm/min scanning speed (Figure 6). The linked pore structure with a 500 μm pore size is present in the PLLA/Mg composite scaffolds made using this method, which is advantageous for cellular activity [61].

### 5.3. Low-Temperature Deposition Manufacturing

LDM is a low-temperature deposition 3D printing method for 3D porous tissue engineering scaffolds. LDM is used to construct the scaffold layer wise on a platform utilizing a low-temperature chamber, according to the computer-aided design data. As illustrated in Figure 7, the scaffold is then freeze-dried to get rid of the frozen solvent. The LDM process, which liquefies materials without heating them, is a type of green manufacturing [67].

Phase separation during the fabrication of the tissue-engineered scaffold by LDM resulted in interconnected microspores in the deposited lines [144]. Compared with FDM and SLS, LDM has the advantage of a non-heating method that preserves the bioactivity of natural biopolymers, such as collagen type I (COLI), gelatin, sodium alginate, and chitosan [144,145,146]. Not only the natural biopolymer but also synthetic biopolymers, such as poly(lactic-*co*-glycolic acid) (PLGA), polyurethane (PU), poly(d,l-lactide) (PDLLA), and poly(l-lactic acid) (PLLA), can also be processed using LDM [147,148]. LDM has the freedom of incorporating inorganic nanoparticles into the biopolymer before processing, which can greatly help improve the scaffolds’ mechanical, physical and biological properties [149,150]. Similarly, Bai et al. used LDM technology to create innovative porous PLGA/TCP/Mg scaffolds using Mg powder, poly (lactide-co-glycolide) (PLGA), and β-tricalcium phosphate (TCP). In vitro analysis was performed on the physical characteristics of the PTM scaffold and the release of Mg ions. The established steroid-associated osteonecrosis (SAON) rabbit model was used to evaluate the osteogenic and angiogenic capabilities of PTM scaffolds and the biosafety following implantation. Their findings demonstrated that the PTM scaffold has superior mechanical properties and a well-designed biomimetic structure [151]. To create a homogeneous ink, metal and polymer particles are dissolved in an organic solvent and thoroughly combined. The solvent 1,4-dioxane has been used extensively, but alternative options include water/ice/ethanol systems [152,153]. Composite scaffolds are printed in a chamber that is roughly −30 °C in temperature. To remove the solvent, lyophilization is performed in the end (Figure 7). The polymer/metal composite scaffolds created using this method had excellent pore connectivity and 45–55% porosity, with macropore and micropore sizes of 400–450 and 5–50 m, respectively [77]. These advantages make the LDM process very suitable for polymer and polymer/metal composite for making scaffolds.

## 6. Conclusions, Challenges, and Future Directions

Although the incorporation of magnesium particles into the polymer matrix offers good biomechanical and biological properties to scaffolds, numerous difficulties pose significant barriers to their widespread adoption. One of the main obstacles in this sector is the poor dispersion of the metallic particles into the polymer matrix. Second, the strength of the scaffolds is substantially degraded by the agglomeration of Mg particles. Additionally, poor chemical and mechanical uniformity of scaffolds and, finally, weak interfacial bonding between Mg and polymers are some of the challenges. Further research is needed for a better polymer/Mg interface to overcome the challenges associated with it. In this regard, surface modification of Mg particles before mixing with the polymer and incorporation of the binder can enhance the surface properties of Mg particles. The biodegradable and biomechanical properties of polymer/Mg composites for biomedical applications can further be enhanced by coatings. The coating is a beneficial technique for surface protection and property improvement as it can generate a nanostructured surface on the top with various pore sizes, which can be helpful for cell attachment [154]. Different Mg alloys need to be investigated in biodegradable polymers for future use.

It is also noteworthy that during the 3D printing process, precise control of printing factors, such as printing speed and nozzle temperature, determines the scaffolds’ characteristics. Additionally, as surface treatment techniques affect scaffolds’ biomechanical and biodegradation properties, they can improve the layer intersection and bonding quality of 3DP scaffolds.

Polymer matrix composites offer so many advantages for scaffold fabrications using 3DP. The capacity of 3DP technology to create complicated porous structures with excellent resolution makes it the most suited method for creating 3D porous scaffolds made of polymer/Mg composites. FDM, LDM, and SLS are the standard techniques used for biodegradable polymer processing. In these procedures, polymer and metal powders are first combined as starting materials. The 3D porous scaffolds are printed (using FDM and LMD) or laser-treated (using SLS) layer by layer. These methods provide polymer/Mg composite scaffolds with relatively high porosity, large pore size, and improved mechanical characteristics, such as compressive strength, modulus, and elongation. The Mg particle’s ability to support loads is the primary cause of the strengthening effect.

Mg particles enhance the biodegradation rate of a biodegradable polymer by changing the wettability of the polymer and by increasing the pH of the degradation medium. This is attributed to the production of alkaline products from the degradation of Mg. These results are clear evidence of the great potential of polymer/Mg composites to be used for biomedical applications.

The authors firmly believe that using a 3D printing technique for polymer/Mg-based composites, is a promising technology for biomedical applications. Despite the low cost, ease of fabrication of complex structures, and rising production rate of polymers, the mechanical properties, and biodegradation of polymer constructs frequently fall short of expectations in matching the original tissues. Therefore, a growing potential for combining Mg with polymers to enhance their mechanical and biological properties using the advanced 3D printing techniques will be in high demand with the increasing demand for implants.

## Figures and Tables

**Figure 1 polymers-14-05460-f001:**
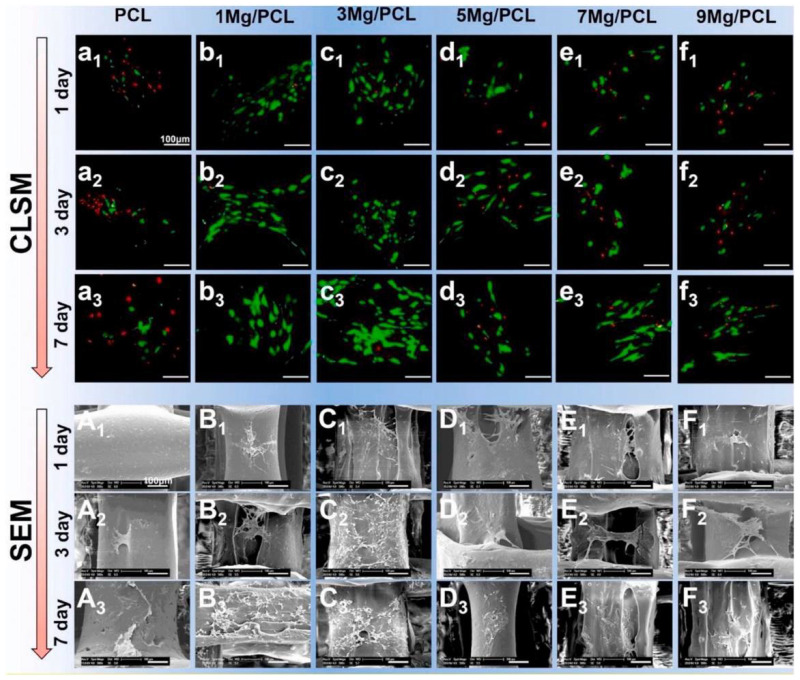
Cell adhesion and proliferation on 3D-printed PCL/Mg composite scaffolds (**a**–**f**) and SEM images of PLC/MG composite with different amounts of Mg (Mg = 0, 1, 3, 5, 7, and 9 wt %) (**A**–**F**) [53].

**Figure 2 polymers-14-05460-f002:**
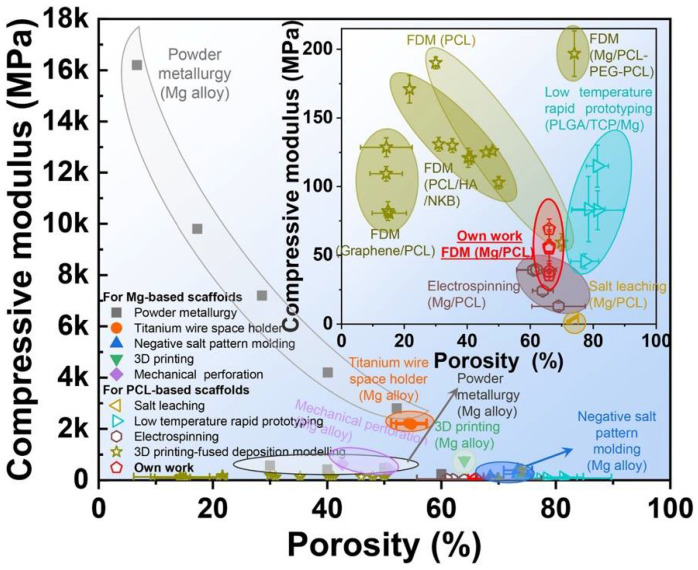
The compressive modulus of Mg-based and PCL-based scaffolds. Figure from [53].

**Figure 3 polymers-14-05460-f003:**
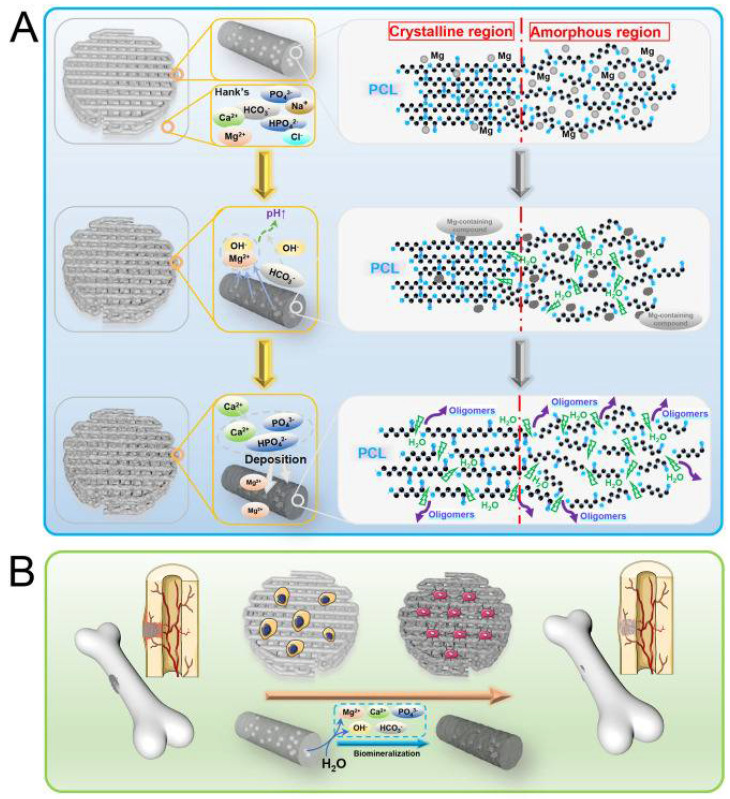
Reactions of Mg/PCL scaffolds for bone healing in vitro and in vivo, showing (**A**) degradation behavior and (**B**) biological activity [53].

**Figure 5 polymers-14-05460-f005:**
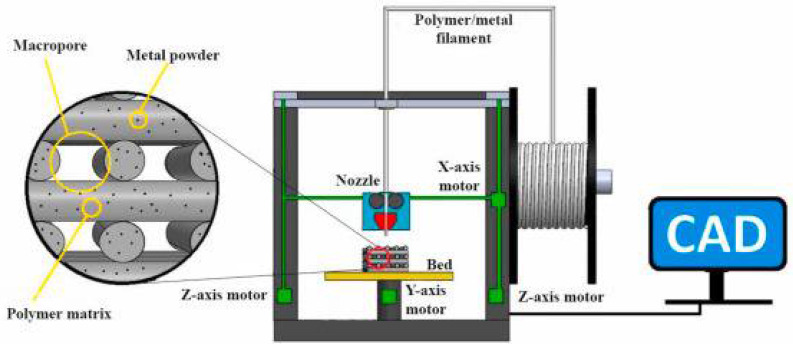
Schematic diagram of the FDM 3D printing process for polymer/metal scaffolds.

**Figure 6 polymers-14-05460-f006:**
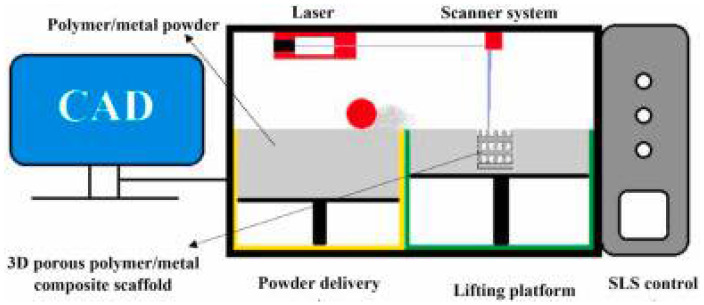
Schematic diagram of the SLS 3D printing technique for polymer/metal scaffolds.

**Figure 7 polymers-14-05460-f007:**
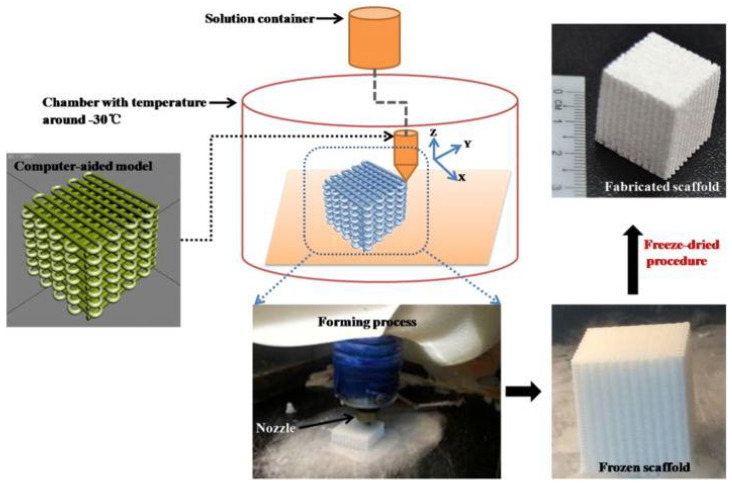
Schematic illustration of LDM 3D manufacturing of scaffold [67].

**Table 1 polymers-14-05460-t001:** Polymer/Mg composite prepared by FDM 3D printing.

Composite	Metal Powder Size (μm)	Nozzle Temperature (°C)	Bed Temperature (°C)	Printing Speed (mm/s)	Reference
PLA/Mg	<50	155	55	40	[135]
PLA/Mg	29.1–64.4	170	60	5	[136]
PCL/Mg	26.8	160	–	1.5	[53]
PCL/Mg	45	110	–	6–8	[52]
PLA/nMgO	0.02	-	180	-	[137]
PLLA/gMgOs	-	-	25		[138]
PLA/Mg	100	-	-	-	[139]

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
