# Peer review of "Additive Manufacturing of Polymer/Mg-Based Composites for Porous Tissue Scaffolds"

_polymers, 2022, doi:10.3390/polym14245460_

Round 1
Reviewer 1 Report
Respected Authors,
It is a pleasure to accept the task of reviewing your Manuscript ID polymers-2064091, entitled “Additive Manufacturing of Polymer/Mg-based Porous Tissue Scaffolds”. In this paper, the authors incorporated the current status and challenges of polymer/Mg composite scaffolds. Additionally, the review also highlights the potential use of 3D printing,
The overall quality of the review is quite good, however, there are shortcomings that should be rectified before acceptance.
Please check the attached file containing my comments (annotated file).

Author Response
Response to Reviewer 1 Comments
We appreciate you and the reviewers for your precious time in reviewing our paper and providing valuable comments. It was your valuable and insightful comments that led to possible improvements in the current version. The authors have carefully considered the comments and tried our best to address every one of them. We hope the manuscript after careful revisions meet your high standards. The authors welcome further constructive comments if any. Many sentences of the manuscript have been carefully rewritten or reorganized to enhance the logic flow and make the statements stricter in a proper tone. We have also included some more relevant and latest literature as recommended by the reviewer. The changes made are highlighted in yellow.
Below we provide the point-by-point responses.
Point 1: Please rewrite this ambiguous sentence.
Response 1: As recommended the sentence is simplified by splitting it into two sentences which is easier to read and understand.
Point 2: Please check and revise for typos mistakes. There are a lot of grametical mistakes which require perusal of the authors. Even abstract contains sentence structuring issues.
Response 2: Thank you for highlighting the issue, we have carefully considered the comment and corrected the typos/mistakes we made in the original manuscript.
Point 3: This review article is missing some latest references related topolymers and tissue engineering. Authores are advised toe incorporate these references provided at various place which greatly enhances the quality of the manuscript.
Response 3: As recommended by the reviewer, the mentioned literature has been added to the manuscript which was much needed and it has added to the quality of this work. We had added the mentioned publibcation into the manuscript and highlighted the changes.
Point 4: The introduction sections more focused towards hard tissues (bone). Therefore, title should contain keyword related to bone. Else rewrite the introduction section..
Response 4: As recommended, the title of the paper is modified.
Point 5: Also natural biopolymers like chitosan can be applied for developing scaffolds, cite this and extract suitable information related to natural biopolymers:
Response 5: The mentioned paper has been cited and the releveant litrature is extracted and the changes are highlighted in the manuscript.
Point 6: Define novelty of your review, which is missing in the current version.
Response 6: The novelty of this work is added to the end of introduction part.
Point 7: The authors have not discussed all conventional techniques. Some are missing, it would be better to write mostly used techniques for developing scaffolds.
Response 7: Thank you for highlighting this, we have added some more techniques which were missing in the original manuscript and the changes have been highlight in the revised manuscript.
Point 8: The current form of this section is not extraordinary. This section has to be thought-provoking. The authors are needed to improve this section.
Response 8: Thank you again for mentioning it. The part has been reviewed and changes have been made and highlighted.

Reviewer 2 Report
The abstract is very poorly written. Almost all of it is a general overview of the use of polymer implants and their comparison with metal samples. In the abstract, innovations and different prominent parts of the article should be presented.
In general, the introduction should be rewritten. Most of this section is general and obvious information.
Page 1
The statistical and general information provided in the first paragraph is not suitable for this article and more specialized references should be used.
Page 2
In the second paragraph, the same procedure has been followed and a series of general and general information has been used, which is not suitable for a specialized article.
References should be numbered consecutively. Reference numbers should be merged within the text.
The term Mg/polymer requires the use and description of more biopolymers. While in this manuscript only PCL and PLA are mentioned. PETG, PMMA, TPU, TPS, and a combination of these materials are included in this category.
Different parts of the paper are written very superficially. For example, for 3D printing, pre-processing is needed to fabricate resin and polymer filament. Many parameters are involved in this process. For example, the distribution of Mg in the polymer matrix.
Table1 on page 11 mentions only three papers. Also, this table does not have any information about mechanical properties.
It is recommended to use the following sources.
https://doi.org/10.1016/j.jmrt.2022.04.076
https://doi.org/10.1016/j.carbpol.2018.11.077
https://doi.org/10.1016/j.eurpolymj.2019.109340
https://doi.org/10.1016/j.mfglet.2022.05.002
https://doi.org/10.1016/j.jmrt.2022.11.024
https://doi.org/10.1016/B978-0-12-819240-5.00012-2
https://doi.org/10.1016/j.addma.2018.03.016
One of the main tasks in the review paper is to categorize and compare materials, methods, and final properties, which are not addressed in this manuscript, and each is presented only in summary form. The following items are suggested to correct the article.
Presentation of biopolymers in a section under the heading Material
Classification of geometrical and material parameters of magnesium on mechanical, biological, decomposition and degradability properties
Providing microstructure, mechanical and biological properties and providing solutions to improve them
Presenting potential and actual applications for bio Mg-polymer structures
Author Response
Response to Reviewer 2 Comments
We appreciate you and the reviewers for your precious time in reviewing our paper and providing valuable comments. It was your valuable and insightful comments that led to possible improvements in the current version. The authors have carefully considered the comments and tried our best to address every one of them. We hope the manuscript after careful revisions meet your high standards. The authors welcome further constructive comments if any. Many sentences of the manuscript have been carefully rewritten or reorganized to enhance the logic flow and make the statements stricter in a proper tone. We have also included some more relevant and latest literature as recommended by the reviewer. The changes made are highlighted in green.
Below we provide the point-by-point responses.
Point 1: The abstract is very poorly written. Almost all of it is a general overview of the use of polymer implants and their comparison with metal samples. In the abstract, innovations and different prominent parts of the article should be presented.
Response 1: As recommended by the reviewer, changes have been made to the abstract.
Point 2: In general, the introduction should be rewritten. Most of this section is general and obvious information.
Response 2: As recommended, the introduction is started with more specific information and more references have been added to the introduction. The changes made are highlighted.
Point 3: The statistical and general information provided in the first paragraph is not suitable for this article and more specialized references should be used.
Response 3: Thank you very much for your comment. The purpose of this section is to give more emphasis on the importance of tissue engineering in orthopedics, that’s why it is included into the manuscript. Still I have modified it by putting more specified references as recommended by the reviewer.
Point 4: In the second paragraph, the same procedure has been followed and a series of general and general information has been used, which is not suitable for a specialized article.
Response 4: The purpose of this section is to give an overview of the current implant materials for permanent implants.
Point 5: References should be numbered consecutively. Reference numbers should be merged within the text.
Response 5: The changes have been made as recommended by the reviewer.
Point 6: The term Mg/polymer requires the use and description of more biopolymers. While in this manuscript only PCL and PLA are mentioned. PETG, PMMA, TPU, TPS, and a combination of these materials are included in this category.
Response 6: The focus of this paper is on the polymer/Mg composites that’s why we have not included other polymer composites such as PLA/TPU etc.
Point 7: Different parts of the paper are written very superficially. For example, for 3D printing, pre-processing is needed to fabricate resin and polymer filament. Many parameters are involved in this process. For example, the distribution of Mg in the polymer matrix.
Response 7: Thank you for highlighting this, we have added the changes to the original manuscript and the changes have been highlight in the revised manuscript.
Point 8: It is recommended to use the following sources.
Response 8: Thank you again for mentioning it. The mentioned references have been added to the manuscript.
Point 9: One of the main tasks in the review paper is to categorize and compare materials, methods, and final properties, which are not addressed in this manuscript, and each is presented only in summary form. The following items are suggested to correct the article.
- Presentation of biopolymers in a section under the heading Material
- Classification of geometrical and material parameters of magnesium on mechanical, biological, decomposition and degradability properties
- Providing microstructure, mechanical and biological properties and providing solutions to improve them
- Presenting potential and actual applications for bio Mg-polymer structures
Response 9: As recommended by the reviewer the effect of various experimental parameters and reinforcement particle size, shape and distribution is added to the 3D printed part of the manuscript and some more literature is cited as recommended by the reviewer.

Round 2
Reviewer 1 Report
This review is in acceptable form, now.
Author Response
No comments by the reviewer on the revised manuscript.
Reviewer 2 Report
Regarding the distribution of magnesium particles and their effect on print quality, the SEM images of literature should be used.
In Table 1, more papers can be presented and reviewed.
Reference number 128 is a duplicate. It is suggested to move with the following reference.
https://doi.org/10.1002/adem.202201309
The quality of the used figures is fantastic. Also, the rest of the requested items have been answered almost completely.
Author Response
All the comments raised by the reviewers are addressed in the revised manuscript.
Also, the English revision has been done for the manuscript and the changes are marked up.
We highly appreciate your recommendations.
